# Low Self-Rated Health as A Risk Factor for Depression in South Korea: A Survey of Young Males and Females

**DOI:** 10.3390/healthcare9040452

**Published:** 2021-04-12

**Authors:** Yunyoung Kim, Eunsu Jang

**Affiliations:** 1Department of Nursing, Andong National University, Andong 36729, Korea; kyy5705@gmail.com; 2Department of Korean Medicine, Daejeon University, Daejeon 34520, Korea

**Keywords:** self-rated health, risk factor, depression

## Abstract

This is a cross-sectional study aimed to examine whether low self-rated health (SRH) is a risk factor for depression among young males and females. Data from the SRH, quality of life (QoL) and depression questionnaires as well as general information, were collected from 512 males and females aged 20–29 years in South Korea. Chi-square test was used to analyze the distribution of depression between the high and low SRH groups. Logistic regression was used to analyze the relationship between SRH and depression after adjusting for covariates. There were 32.6% males and 30.1% females who were at risk for depression. A significant difference in the distribution of depression between the low and high SRH groups in both males and females was found. The low SRH group had a higher prevalence of depression than the high SRH group in both males and females in the crude analysis. However, when the BMI, economic status, and mental component score were adjusted, the OR of the low SRH group was still significant in males. Low SRH may be a risk factor for depression especially in males. Further studies to improve SRH by developing preventive measures against depression status while considering gender characteristics are needed.

## 1. Introduction

South Korea has experienced a variety of sociocultural upheavals, such as family dissolution, increase in the temporary workforce, and the aging of society, which are derived from its rapid economic growth. Psychological pressure resulting from this phenomenon is considered one of the important reasons for the increasing prevalence of depression [1]. According to the Health Insurance Review and Assessment Service in Korea, the number of patients diagnosed with depression increased to 682,000 in 2015 from 602,000 in 2011 [2]. It is also estimated that there are more than 2 million patients with depression who need medical care but refuse it due to the stigma of depression [3].

The increasing prevalence of depression can cause socioeconomic loss, and the medical expense for depression increased from KRW 73.9 billion in 2011 to KRW 91.5 billion in 2015 [2]. Furthermore, the World Health Organization predicted that depression will be ranked second in the economic burden of disease areas by 2020 [4]. Along with the increase in medical expense, the indirect socioeconomic loss associated with depression encompasses physical, social, and functional disabilities that may result in dropping out of school, a decrease in income, job loss, and even suicide. Depression has been correlated with demographic factors such as economic status, residence type, smoking, drinking, and body mass index (BMI) [5,6,7]. This means that these factors could affect the increase or decrease in depression. Furthermore, depression during young adulthood can increase the risk of not only chronic depression but also other major mental disorders [8,9], especially since the recent increase of depression in young adults aged 20–29 was higher than that of other generations [10]. It is also known that females are more vulnerable to depression than males because of ovarian hormones; estrogen and progesterone, deteriorate response to stress, female’s lack of social power to specific major traumas, particularly sexual abuse, and so on [11]. However, male depression is also a serious concern [12].

It is also known that females are more vulnerable to depression than males [11]. However, male depression is also a serious concern [12].

Self-rated health (SRH), an individual’s self-perceived health status, including his/her thinking, as well as the quality of life (QoL) are related with each other [13] and they are associated with depression [14]. SRH has been a significant predictor of various aspects of current health status [15]. In addition to its relationship with current health status, several studies have shown that SRH is closely related to mortality, successful aging, and life satisfaction [16,17,18,19].

As the prevalence of depression is increasing in South Korea, many research studies of depression are being conducted nationwide. The Korea National Health and Nutrition Examination Survey (KNHANES) has investigated the association between SRH and depression and revealed that SRH is considered an important indicator of depression [20,21,22]. Several studies insisted that the standardized depression tool used in KNHANES, the Patients Health Questionnaire-9, had limitations for indicating detailed symptoms of depression [22,23]. The Center for Epidemiologic Studies of Depression (CES-D) scale is a short self-report questionnaire designed to measure depressive symptomatology in the general population and has sensitivity to detect various aspects of depression.

This study aimed to investigate depression using the CES-D scale and to elucidate whether SRH is an independent risk factor for depression in males and females aged 20–29 years.

## 2. Materials and Methods

### 2.1. Participants and Data Collection Procedures

This study was a cross-sectional study, and data were collected by convenience sampling of males and females aged 20–29 years who were residents of Andong, Daejeon, and Cheonan cities in South Korea from October 2020 to December 2020.

The researcher publicized and fully explained the purpose of this study to young males and females around the universities of three cities. People who showed interest in this study were included. However, we excluded several individuals, which could affect the hypothesis of this study; people who showed psychotic symptoms or had a history of psychotic symptoms; people who had experienced a manic episode, hypomanic episode, or mixed episode; people who had been dependent on alcohol or other substances; and/or people who were given an anxiolytic, antidepressant, antipsychotic, corticosteroid, female hormone, L-dopa, digitalis, bromide, cyclosporin, disulfiram, isoniazid, or yohimbine (substances that may affect depression) within two weeks of participation were excluded.

The participants voluntarily completed the questionnaire after providing informed consent. A total of 550 questionnaires were collected from the participants, and a total of 512 questionnaires (excluding 38 questionnaires with missing responses) were divided by gender and depression/non-depression groups and used in the final analysis (Figure 1).

This study was approved by the institutional review board of Andong University (Approval No; 1040191-202010-HR-017-01).

### 2.2. Measurements

The questionnaire contained seven items about general characteristics (age, height, weight, residence type, economic status, drinking, and smoking), one item about the SRH status, 20 items of a depression measurement tool, and 12 items about QoL.

#### 2.2.1. Self-Rated Health

The average health status was rated on a Likert scale with the responses of following response options: “the best”, “very good”, “good”, “somewhat bad”, and “bad”. The SRH status was defined as good for the answers of “the best”, “very good”, and “good”, and bad for the answers of “somewhat bad” and “bad”. Accordingly, participants who had “good” answers were categorized in the high SRH group and those who had “bad” answers were categorized in the low SRH group.

#### 2.2.2. Depression

To evaluate the degree of the participants’ depression, the CES-D scale, which was developed by Radloff [24] and then modified by Cho et al. [25], was used. The CES-D scale is a self-report questionnaire that is composed of 20 questions that measure depressive symptoms experienced during the past week. The participants rated each question as “extremely rare” (0 points), “occasionally” (1 point), “often” (2 points), and “mostly” (3 points). Participants who scored less than 16 points, which is the cut-off point suggested by Radloff, were classified as the non-depressive group and those who scored equal to or greater than 16 points were classified as the depressive group. The internal consistency of the CES-D was Cronbach’s α = 0.85 at the time of development, and Cronbach’s α = 0.91 in this study.

#### 2.2.3. QoL

To measure the participants’ QoL, the Short Form-12 Health Survey Questionnaire (SF-12) was used. The SF-12 is a short version of the Short Form-36 Health Survey Questionnaire by Ware and Sherboume [26], and it is composed of 12 questions. It contains a physical component score (PCS) and a mental component score (MCS). The use of the SF-12 was permitted by US QualityMetric Inc., and their calculation method was used to obtain the QoL score; a higher QoL score indicates a higher QoL. The internal consistency of PCS and MCS of the QoL was Cronbach’s α = 0.85 and 0.76 at the time of development and Cronbach’s α = 0.76 and 0.77 in this study, respectively.

### 2.3. Statistical Analysis

In the sample size of this study, we referred to a previous study [27]. The number of subjects was calculated using the G-power 3.1.9.4 program. To perform logistic regression analysis, the sample size was calculated with Odds Ratio = 1.45, probability = 0.02, and power = 0.95. As a result, the minimum sample size was 496.

The collected data were analyzed using the IBM SPSS 24.0 statistics program (IBM Corp., Armonk, NY, USA). A frequency analysis was performed for the participants’ general characteristics and the differences in their general characteristics according to SRH status were analyzed using the *t*-test and chi-square test. Differences in the distributions of SRH status between the depressive and non-depressive groups were analyzed using the chi-square test. The relative risk of depression according to SRH status was analyzed using logistic regression. The analysis was performed by dividing the model by age, type of housing, and smoking, which showed differences between the groups. The accepted significance level was *p* < 0.05.

## 3. Results

### 3.1. General Characteristics of The Participants

Regarding the general characteristics of the male participants, 153 (67.4%) had a depression score under 16, and 74 (32.6%) had a depression score of 16 or more. For the female participants, 197 (69.1%) had a depression score under 16, and 88 (30.1%) had a depression score of 16 or more. For the male participants, there was no difference in age, BMI, residence type, drinking, or smoking; there was a difference in economic status (*p* = 0.007), PCS (*p* = 0.003) and MCS (*p* < 0.001). For the female participants, there was no difference in age, residence type, economic status, smoking or drinking habits or PCS, but there was a difference in BMI (*p* = 0.010) and MCS (*p <* 0.001) (Table 1).

### 3.2. Differences in The Distribution of SRH Status between The Depression Groups

Of the male participants with a high SRH status, there were 117 (77.5%) in the non-depressive group and 34 (22.5%) in the depressive group of the male participants with a low SRH status, there were 36 (47.4%) in the non-depressive group and 40 (52.6%) in the depressive group. There was a significant difference in the distribution of depression (*p* < 0.001). For the female participants with a high SRH status, 164 (75.9%) were in the non-depressive group and 52 (24.1%) were in the depressive group of the female participants with a low SRH status, 33 (47.8%) were in the non-depressive group and 36 (52.2%) were in the depressive group. There was a significant difference in the distribution of depression (*p* < 0.001; Table 2).

### 3.3. Relative Risk of Depression According to The Low and High SRH Groups

When the relative risk of depression was assessed by dividing the male and female participants into the high SRH group and low SRH groups, the crude odds ratio (OR) of the low SRH group compared to the high SRH group was 3.824 (2.119–6.900, *p* < 0.001) in males and 3.441 (1.953–6.060, *p* < 0.001) in females.

When the BMI, which showed a significant difference in the general characteristics, was adjusted, the OR of the low SRH group was 3.805 (2.107–6.871, *p* < 0.001) in males and 3.297 (1.858–5.849, *p* < 0.001) in females. When the BMI and economic status were adjusted, the OR of the low SRH group was 3.693 (1.999–6.627, *p* < 0.001) in males and 3.299 (1.858–5.858, *p* < 0.001) in females. When the BMI, economic status, and PCS were adjusted, the OR of the low SRH group was 3.123 (1.553–6.280, *p* < 0.001) in males and 3.616 (1.733–7.544, *p* < 0.001) in females, respectively, showing that there was a consistent significant difference.

However, when the BMI, economic status, and MCS were adjusted, the OR of the low SRH group was 2.629 (1.344–5.413, *p* < 0.01) in males and 1.883 (0.947–3.744, *p >* 0.05) in females (Table 3).

## 4. Discussion

According to research into the mental health of South Korean individuals aged 20–29 years, 26.8% of them have depression [28]. This indicates that South Korean individuals aged 20–29 years have a high risk of depression, considering the 8–18% prevalence of depression among the nation’s general adult population. In case of Turkey, the depression prevalence of young adult was 29%, which is similar to South Korea [29].

Many previous studies have focused on influential factors such as stress, self-efficacy, and drinking, which are directly and indirectly related to depression [30,31,32]. In addition to these factors, SRH is gaining attention due to its importance in predicting depression [15]. Since SRH encompasses all areas of health, including physical, cognitive, and mental health along with the concept of well-being, it is believed that SRH can be improved with various coordinated efforts [33]. Therefore, it is very important to determine whether SRH acts as a risk factor for depression, which can decrease QoL and prevent the performance of social roles [34].

In this study, although there was a gender difference, the SRH of the males and aged 20–29 years was affected by BMI, economic status, PCS and MCS.

This study revealed that there was a significant difference in the distribution of depression in males according to economic status. This meant that low income may cause depressive mood and affect depression. Previous study also suggested that employment status and the ratio of debts to assets were significantly associated with depression [7]. High economic state increases confidence of people, and low economic income is recognized as a major risk factor for depression [35,36]. From this, we can insist that socioeconomic deprivation is significantly a positive relationship with depression, and this effect is growing as the age increases. Therefore, it is important to prevent early depression from the young generation actively.

This study also revealed that there were significant differences in PCS and MCS in males and MCS in females between the depressive group and the non-depressive group. We think that depressed people usually had a lower QoL, especially MCS. A previous study supports our result that depressed patients have QOL deficits that are directly attributable to their mood disturbance [37].

In terms of BMI, there was a significant difference among the females, and the BMI of the depressive group was lower than that of the non-depressive group. This indicates that BMI may possibly be a risk factor for depression. Carolyn et al. also revealed that BMI was associated with depression in females and suggested BMI as a preventive factor for depression [5].

We classified the participants into two groups using CES-D Score and SRH. The results showed that there were fewer participants with depression in the high SRH group for both males and females. The KNHANES study in 2011 reported that people tend to think their SRH is worse if they have more depressive symptoms and other studies have also shown that people have a more advanced level of depression when they have worse SRH [38,39].

In this study, we observed that both females and males who had a low SRH score tended to have a lower score for physical and mental QoL. As people perceive that they are in bad health, they are more likely to have negative thinking, and to, therefore, be unsatisfied with their life, which decreases their QoL. A study also reported a positive correlation between SRH and life satisfaction [17].

SRH is a crucial factor for determining both physical and mental QoL [18]. Thus, a negative perception of an individual’s own health can result in negative influences on his/her entire life [40]. This means a positive perception of the health can be beneficial to an individual’s life.

This study revealed that both females and males in the low SRH group showed a higher risk of depression. This tendency was still significant after adjusting for BMI, economic status, and PCS in males. This is similar to Cho’s research that mental health is significantly associated with low QoL in depressive individuals [41]. However, while the males in the low SRH group also showed significant differences after adjusting for BMI, economic status, and MCS, females in the low SRH group did not show significant differences after adjusting for BMI, economic status, and MCS. This revealed that low SRH affected depression regardless of BMI, economic status, and MCS in males, while this was not the case in females.

From the results mentioned above, we can infer that low SRH is an independent risk factor for depression. It means that if we may turn low SRH into high one, the prevalence of depression can be decreased in the young generation.

Therefore, further studies are needed to investigate strategies for improving SRH considering the difference in gender characteristics. We hope that studies and program development for improving low SRH will be actively conducted and continued in the future.

However, it is also possible that early depression may have affected the young males and females’ self-health ratings, and result in low health perception, on the contrary. It should be demonstrated in the further study.

This study, which aimed to elucidate the relationship between SRH and depression, shows that low SRH can be an independent risk factor for depression. Additionally, low SRH might be closely associated with or interact with MCS in females. This result is remarkable in that improving SRH may help to prevent depression. However, this study has several limitations: it only used a survey tool rather than expert assessments of the participants’ depression for a true depression diagnosis, and this should be considered when generalizing the results. This study also collected limited information on general demographics and basic lifestyle information. Nevertheless, SRH, sleep quality, and job stress seem to be correlated and should be considered in future research.

## 5. Conclusions

The purpose of this study was to investigate the relationship between SRH status and depression in young adulthood. The low SRH group had a higher prevalence of depressive symptoms than the high SRH group, which was significant even after adjusting for general demographic differences. Therefore, a low SRH status can be considered an independent risk factor for depression especially in males. However, MCS may be associated or interact with depression in females. We hope that the study’s results can be used as the basis for developing ways to improve SRH status while considering gender characteristics in the future.

## Figures and Tables

**Figure 1 healthcare-09-00452-f001:**
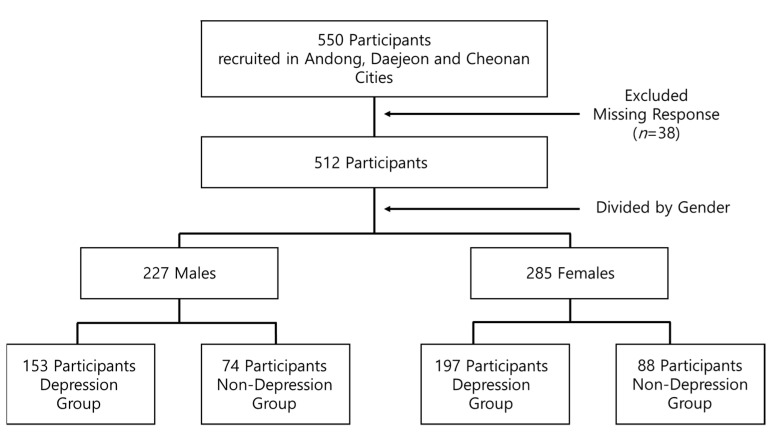
The flow diagram of participants.

**Table 1 healthcare-09-00452-t001:** General characteristics of the participants (*n* = 512).

Variables		Males (*n* = 227)			Females (*n* = 285)		
		Depression	Depression	t/χ2 (*p*)	Depression	Depression	t/χ2 (*p*)
		Score	Score		Score	Score	
		<16	≥16		<16	≥16	
		*n* (%) M ± SD	*n* (%) M ± SD		*n* (%) M ± SD	*n* (%) M ± SD	
Age		23.2 ± 4.33	23.0 ± 3.68	0.468 (0.640)	21.0 ± 1.96	22.0 ± 4.30	−1.929 (0.057)
BMI		23.4 ± 3.02	23.1 ± 2.77	0.770 (0.442)	20.2 ± 2.50	21.1 ± 3.09	−2.607 (0.010)
Residence Type	Own Home	30 (19.6)	16 (21.6)	0.639	67 (34.0)	26 (29.5)	5.676
	Rental	101 (66.0)	45 (60.8)	(0.727)	68 (34.5)	43 (48.9)	(0.059)
	Other	22 (14.4)	13 (17.6)		62 (31.5)	19 (21.6)	
Economic Status	Good	111 (72.5)	40 (54.1)	7.661	159 (80.7)	73 (83.0)	0.2020
	Poor	42 (27.5)	34 (45.9)	(0.007)	38 (19.3)	15 (17.0)	(0.393)
Drinking	Yes	81 (52.9)	41 (55.4)	0.122	89 (45.2)	39 (44.3)	0.018
	No	72 (47.1)	33 (44.6)	(0.418)	108 (54.8)	49 (55.7)	(0.498)
Smoking	Smoker	34 (22.2)	23 (31.1)	2.082	7 (3.6)	3 (3.4)	0.004
	Nonsmoker	119 (77.8)	51 (68.9)	(0.101)	190 (96.4)	85 (96.6)	(0.627)
PCS		52.7 ± 6.00	49.5 ± 7.97	3.045 (0.003)	52.8 ± 5.95	51.4 ± 9.11	1.325 (0.188)
MCS		47.4 ± 8.60	37.4 ± 8.35	8.235 (<0.001)	47.6 ± 7.11	37.0 ± 8.27	11.068 (<0.001)

BMI, body mass index; PCS, physical component score; MCS, mental component score.

**Table 2 healthcare-09-00452-t002:** The prevalence of depression according to self-rated health group.

						(*n* = 512)
Variables	Men (*n* = 227)		χ2 (*p*)	Women (*n* = 285)		χ2 (*p*)
	Depression	Depression		Depression	Depression	
	Score	Score		Score	Score	
	<16	≥16		<16	≥16	
	*N* (%)	*N* (%)		*N* (%)	*N* (%)	
SRH High Group	117 (77.5)	34 (22.5)	20.867 (<0.001)	164 (75.9)	52 (24.1)	19.347 (<0.001)
SRH Low Group	36 (47.4)	40 (52.6)		33 (47.8)	36 (52.2)	

SRH (self-rated health).

**Table 3 healthcare-09-00452-t003:** Adjusted ORs (95% CI) for depression by self-rated health assessment group.

			(*n* = 512)	
	Men (*n* = 227)		Women (*n* = 285)	
	SRH High Group	SRH Low Group	SRH High Group	SRH Low Group
Depression		ORs (95% CI)		ORs (95% CI)
Crude	Reference	3.824 (2.119–6.900) ***	Reference	3.441 (1.953–6.060) ***
Model 1	Reference	3.805 (2.107–6.871) ***	Reference	3.297 (1.858–5.849) ***
Model 2	Reference	3.693 (1.999–6.627) ***	Reference	3.299 (1.858–5.858) ***
Model 3	Reference	3.123 (1.553–6.280) **	Reference	3.616 (1.733–7.544) **
Model 4	Reference	2.629 (1.344–5.413) **	Reference	1.883 (0.947–3.744)

SRH (self-rated health); ORs, odds ratios; CI, confidence interval. Model 1. BMI; Model 2. BMI, economic status; Model 3. BMI, economic status, PCS; Model 4. BMI, economic status, MCS, *** *p* < 0.001, ** *p* < 0.01.

## Data Availability

The data presented in this study are available on request from the corresponding author. The data are not publicly available due to privacy or ethical restrictions.

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
