# Peer review of "Low Self-Rated Health as A Risk Factor for Depression in South Korea: A Survey of Young Males and Females"

_healthcare, 2021, doi:10.3390/healthcare9040452_

Round 1

Reviewer 1 Report

Excellent study examining an important question.  One question, might be addressed in the Discussion: 

acknowledgement of the possibility that early depression may have affected the student's self-health ratings, making the case for a persistence of depression and its effect on health perception, not just a predictive/determinative role of poor health in increasing risk of depression.

Author Response

Answer to Reviewer’s Comments

We sincerely thank the reviewer for your effort and time in reviewing our manuscript. A detailed point-by-point rebuttal and list of changes to the revised manuscript are provided below. Your comments have improved the overall quality of the manuscript. All of the authors truly appreciate you for your efforts and help.

First Reviewer’s report:

Excellent study examining an important question. One question, might be addressed in the Discussion: acknowledgement of the possibility that early depression may have affected the student's self-health ratings, making the case for a persistence of depression and its effect on health perception, not just a predictive/determinative role of poor health in increasing risk of depression.

  • In response to the reviewer’s comment, we added the possibility that early depression may have affected the young males and females' self-health ratings (7 page, 247-249 lines).

--> However, it is also possible that early depression may have affected the young males and females' self-health ratings, and result in low health perception, on the contrary. It should be demonstrated in the further study.

We sincerely thank you for your comment which was useful to enhance our manuscript

Reviewer 2 Report

Thank you for the opportunity to review the paper entitled: "low self-rated health as a risk factor for depression in south korea: A survey for young males and females" . This paper adresses and interesting topic and it is well written and the results and interpretation are well described. But I would like to share with the authors some comments:

Introduction section: the authors mentioned that females are more vunerable than males but there depresison is also relevant for males. It could be useful to explain the reasons of these assumptions.

Method section: I strongly recommend to include a new section of procedure in wich you can explain in detail the research process (the dates and places of the research, who did the recruitment process, who collected the data, the ethics committe´s approval, including the reference number if any).

Participants: please include in a more explicit way the inclusion and exclusion criteria of your study. I am not sure about whether you are talking about random or convenience sampling. As it is describe in the text, it is more likely to be convenience sampling, please clarify.

Measurements: Please include the internal consistency of the different instruments.

Data analysis: please specify the model type applied in the regression models. I strongly recommend to follow the corresponding equator network checklist for regression models: https://www.equator-network.org/.  Consider to include a sample size estimation to support your data.

Results: The results are well explained. Line 127, please remove the = symbol.

Discussion: The results are presented in a research context but it is necessay to enchance the interpretation of the results, including an explanation. For example, the authors stated that there is a significant depression in males related with socioeconomic status, but there is no a clear explanation of that.

I hope that this comments will be useful to enhance your manuscript.

Author Response

Second reviewer’s comment

We sincerely thank the reviewer for your effort and time in reviewing our manuscript. A detailed point-by-point rebuttal and list of changes to the revised manuscript are provided below. Your comments have improved the overall quality of the manuscript. All of the authors truly appreciate you for your efforts and help.

  1. Introduction section: the authors mentioned that females are more vunerable than males but there depression is also relevant for males. It could be useful to explain the reasons of these assumptions.

  • There has been several researches that females are more vulnerable in depression than males. Females’ biological state and lack of social power make them more vulnerable than males. In response to the reviewer’s comment, we have now added the reasons why females are more vulnerable than males, (1-2 page, 44-48 lines).

--> It is also known that females are more vulnerable to depression than males because of ovarian hormones; estrogen and progesterone, deteriorate response to stress, female’s lack of social power to specific major traumas, particularly sexual abuse, and so on [11]. However, male depression is also a serious concern [12]

  1. Method section: I strongly recommend to include a new section of procedure in which you can explain in detail the research process (the dates and places of the research, who did the recruitment process, who collected the data, the ethics committe´s approval, including the reference number if any).
  • In response to the reviewer’s comment, we added the research process (the dates and places of the research, who did the recruitment process, who collected the data, the ethics committe´s approval, including the reference number, (2 page, 71-92 lines).

--> This study was a cross-sectional study, and data were collected by convenience sampling of males and females aged 20–29 years who were residents of Andong, Daejeon, and Cheonan cities in South Korea from October 2020 to December 2020. The researcher publicized and fully explained the purpose of this study to young males and females around the universities of three cities. People who showed interest in this study were included. However we excluded several occasions, which could affect to hypothesis of this study; people who showed psychotic symptoms or had a history of psychotic symptoms; people who had experienced a manic episode, hypomanic episode, or mixed episode; people who had been dependent on alcohol or other substances; and/or people who were given an anxiolytic, antidepressant, antipsychotic, corticosteroid, female hormone, L-dopa, digitalis, bromide, cyclosporin, disulfiram, isoniazid, or yohimbine (substances that may affect depression) within two weeks of participation were excluded. The participants voluntarily completed the questionnaire after providing informed consent.

In this study, the number of subjects was calculated using the G-power 3.1.9.4 program. To perform logistic regression analysis, the sample size was calculated with Odds Ratio=1.45, probability=.02, and power=0.95. As a result, the minimum sample size was 496. A total of 550 questionnaires were collected from the participants, and a total of 512 questionnaires (excluding 38 questionnaires with missing responses) were used in the final analysis.

This study was approved by the institutional review board of Andong University (Approval No; 1040191-202010-HR-017-01).

  1. Participants: please include in a more explicit way the inclusion and exclusion criteria of your study. I am not sure about whether you are talking about random or convenience sampling. As it is describe in the text, it is more likely to be convenience sampling, please clarify.
  • In response to the reviewer’s comment, we clarified sampling method (2 page, 71-72 lines).

--> This study was a cross-sectional study, and data were collected by convenience sampling of males and females aged 20–29 years

  1. Measurements: Please include the internal consistency of the different instruments.
  • In response to the reviewer’s comment, we added the internal consistency of the different instruments, (2 page, 112-114 lines) (3 page, 121-123 lines).

--> The internal consistency of the CES-D was Cronbach's α=.85 at the time of development, and Cronbach's α=.91 in this study.

The internal consistency of PCS and MCS of the QoL was Cronbach's α=.85 and .76 at the time of development and Cronbach's α=.76 and .77 in this study, respectively.

  1. Data analysis: please specify the model type applied in the regression models. I strongly recommend to follow the corresponding equator network checklist for regression models: https://www.equator-network.org/. Consider to include a sample size estimation to support your data.
  • In response to the reviewer’s comment, we included sample size estimation to support data. (2 page, 85-90 lines).

--> In this study, the number of subjects was calculated using the G-power 3.1.9.4 program. To perform logistic regression analysis, the sample size was calculated with Odds Ratio=1.45, probability=.02, and power=0.95. As a result, the minimum sample size was 496. A total of 550 questionnaires were collected from the participants, and a total of 512 questionnaires (excluding 38 questionnaires with missing responses) were used in the final analysis.

  1. Results: The results are well explained. Line 127, please remove the = symbol.
  • In response to the reviewer’s comment, we remove the = symbol (3 page, 141 lines)

--> MCS (p <.001).

  1. Discussion: The results are presented in a research context but it is necessay to enchance the interpretation of the results, including an explanation. For example, the authors stated that there is a significant depression in males related with socioeconomic status, but there is no a clear explanation of that.
  • In response to the reviewer’s comment, we have now added details related to the economic status and depression in our manuscript (6 page, 201-205 lines) (6 page, 241-242 lines) (9 page, 346-348 lines).

--> High economic state increases confidence of people, and low economic income is recognized as a major risk factor for depression [34, 35]. From this, we can insist that socioeconomic deprivation is significantly a positive relationship with depression, and this effect is growing as the age increases Therefore it is important to prevent early depression from young generation actively

ref. Pak TY, Choung YJ. Relative deprivation and suicide risk in South Korea. Social Science & Medicine.2020:247;112815

ref. Chang SM, Hong JP, Cho MJ. Economic burden of depression in South Korea. Soc Psychiatry Psychiatr Epidemiol.2012;47:683–689

--> It means that if we may turn low SRH into high one, the prevalence of depression can be decreased in young generation.

We sincerely thank you for your comment which was useful to enhance our manuscript

Round 2

Reviewer 2 Report

Thank you for  the opportunity to review the revised version of this manuscript.

The authors have done a good job considering my previous comments, but I would like to comment a couple of questions that arise when reading the revised manuscript.

  • Sample size estimation fits better in statistical analysis section. Furthermore, the authors must justify why they coose these data in order to perform this sample size estimation. These data comes from a previous similar research? If so, the citation must be included.
  • Following my previous recommendation, I still recommend to the authors to include the equator network checklist for this type of research as supplementary file.

I hope that these comments will be useful.

Author Response

Answer to Reviewer’ Comments

We sincerely thank the reviewer for your effort and time in reviewing our manuscript. A detailed point-by-point rebuttal and list of changes to the revised manuscript are provided below. Your comments have improved the overall quality of the manuscript. All of the authors truly appreciate you for your efforts and help.

Reviewer’s report:

The authors have done a good job considering my previous comments, but I would like to comment a couple of questions that arise when reading the revised manuscript.

Sample size estimation fits better in statistical analysis section. Furthermore, the authors must justify why they coose these data in order to perform this sample size estimation. These data comes from a previous similar research? If so, the citation must be included. Following my previous recommendation, I still recommend to the authors to include the equator network checklist for this type of research as supplementary file.

  1. In response to the reviewer’s comment, we moved sample size estimation to statistical analysis section and attached citation reference. This study considered the sample size of previous study. [Whitton SW, Kuryluk AD. Relationship satisfaction and depressive symptoms in emerging adults: cross-sectional associations and moderating effects of relationship characteristics. J Fam Psychol. 2012;26(2):226-35.. DOI: 10.1037/a0027267]
  2. We also included the equator network checklist for observational study (cross-sectional design) as supplementary file and used a flow diagram in this article (figure 1).

Figure 1. The flow diagram of participants

We really thank you for your comment which was useful to enhance our manuscript.
